# The 4Rs Framework of Sports Nutrition: An Update with Recommendations to Evaluate Allostatic Load in Athletes

**DOI:** 10.3390/life15060867

**Published:** 2025-05-27

**Authors:** Diego A. Bonilla, Jeffrey R. Stout, Michael Gleeson, Bill I. Campbell, Guillermo Escalante, Daniel Rojas-Valverde, Jorge L. Petro, Richard B. Kreider, Adrián Odriozola-Martínez

**Affiliations:** 1Research Division, Dynamical Business & Science Society—DBSS International SAS, Bogotá 110311, Colombia; jorgelpetro@correo.unicordoba.edu.co; 2Hologenomiks Research Group, Department of Genetics, Physical Anthropology and Animal Physiology, University of the Basque Country (UPV/EHU), 48940 Leioa, Spain; adrianodriozola@gmail.com; 3Grupo de Investigación NUTRAL, Facultad Ciencias de la Nutrición y los Alimentos, Universidad CES, Medellín 050021, Colombia; 4Physiology of Work and Exercise Response (POWER) Laboratory, Institute of Exercise Physiology and Rehabilitation Science, University of Central Florida, Orlando, FL 32816, USA; jeffrey.stout@ucf.edu; 5School of Sport, Exercise and Health Sciences, Loughborough University, Loughborough LE11 3TU, UK; m.gleeson@lboro.ac.uk; 6Performance & Physique Enhancement Laboratory, University of South Florida, Tampa, FL 33620, USA; bcampbell@usf.edu; 7Department of Kinesiology, California State University San Bernardino (CSUSB), San Bernardino, CA 92407, USA; gescalan@csusb.edu; 8Centro de Investigación y Diagnóstico en Salud y Deporte (CIDISAD), Escuela Ciencias del Movimiento Humano y Calidad de Vida (CIEMHCAVI), Universidad Nacional, Heredia 86-3000, Costa Rica; drojasv@una.cr; 9Clínica de Lesiones Deportivas (Rehab & Readapt), Escuela Ciencias del Movimiento Humano y Calidad de Vida (CIEMHCAVI), Universidad Nacional, Heredia 86-3000, Costa Rica; 10Grupo de Investigación en Ciencias de la Actividad Física, el Deporte y la Salud (GICAFS), Universidad de Córdoba, Montería 230002, Colombia; 11Exercise & Sport Nutrition Laboratory, Human Clinical Research Facility, Texas A&M University, College Station, TX 77843, USA

**Keywords:** allostasis, sports nutritional sciences, physiological adaptation, biomarkers, cacostasis

## Abstract

The 4Rs of sports nutrition were proposed in recent years as an evidence-based framework to optimize post-exercise recovery within the context of allostasis. Under this paradigm, it is important to consider that each R represents a factor with a tremendous influence on the allostatic response and improves individual components of the allostatic load (AL), which will positively impact the exercise-induced adaptations and the athlete’s recovery. The 4Rs correspond to the following. (i) Rehydration—This is necessary to guarantee the post-exercise consumption of at least 150% of the body mass lost during the exercise accompanied by sodium (if faster replacement is required). (ii) Refuel—Carbohydrate intake (~1.2 g/kg body mass per hour for up to 4 h post-exercise) is essential not only in restoring glycogen reserves but also in supporting the energy needs of the immune system and facilitating tissue repair. Despite changes in substrate utilization, a ketogenic diet generally has neutral or negative effects on athletic performance compared to carbohydrate-rich diets. (iii) Repair—The ingestion of high-quality protein stimulates post-exercise net muscle protein anabolism and might contribute to faster tissue growth and repair. The use of certain supplements, such as creatine monohydrate, might help to enhance recovery, while tart cherry, omega-3 fatty acids, and dietary nitrate (e.g., *Beta vulgaris*, *Amaranthus* L.), as well as other herbal extracts containing flavonoid-rich polyphenols, deserve further clinical research. (iv) Recuperate—Pre-sleep nutrition (casein- or protein-rich meal with slow digestion rate) has a restorative effect, facilitating the recovery of the musculoskeletal, endocrine, immune, and nervous systems. In this article, we update the 4Rs framework, delve deeper into the allostasis paradigm, and offer theoretical foundations and practical recommendations (the 4Rs app) for the assessment of AL in athletes. We cautiously propose an AL index (AL_index_) for physique competitors and elite athletes to evaluate the cumulative physiological stress induced by exercise and, thereby, to adjust exercise and nutrition interventions.

## 1. Introduction to the 4Rs Framework of Sports Nutrition

To better understand the nutritional strategies that influence post-exercise recovery, the 4Rs—Rehydrate, Refuel, Repair, and Rest—was proposed as an operational framework for sports nutrition by Bonilla et al. (2020) [1]. These four Rs do not attempt to replace existing techniques or construct a new, rigid paradigm in this regard, but rather seek to introduce the strategic application of nutritional strategies that should be considered during the recovery process in athletes (Figure 1). In practice, “Recuperate” might be considered a broader term that includes not only passive recovery (e.g., sleep) but also active strategies to address fatigue accumulation, injury recovery, or mental relaxation in order to restore functionality [2]. Overall, the Rest/Recuperate component highlights the importance of downtime to repair tissue and restore energy, including passive and active strategies depending on the athlete’s situation.

This approach divides the nutrition intervention into four interrelated scenarios that follow the post-exercise time course to optimize the exercise-induced adaptations and recovery. Available clinical evidence and recommendations by professional organizations support the structure of the 4Rs framework, as shown in Table 1.

When the strategies outlined in the 4Rs framework are effectively implemented through consistent nutritional adherence and compliance, exercise-induced adaptations may be optimized. In fact, several research groups have recommended or used this approach for the metabolomic and proteomic profiling of athletes [56], as a holistic view of nutrition and healing in injured athletes [57,58], as a working model that should be prioritized to understand physiological requirements and nutritional recommendations for equestrian riders [59], and during the promotion of health-related lifestyles among university students [60].

In this article, we delve into the allostasis paradigm as the foundation of the 4Rs framework and explore recommendations for the evaluation of allostatic load. We suggest an allostatic load index (AL_index_) for physique competitors and elite athletes to assess the cumulative physiological stress induced by exercise.

## 2. Materials and Methods

This review incorporated elements of the Preferred Reporting Items for Systematic Reviews and Meta-Analyses Extension for Scoping Reviews (PRISMA-ScR) [61]. It encompasses the identification, selection, evaluation, and synthesis of available evidence in a narrative format.

### 2.1. Information Sources

The primary sources for the articles included the following online databases: PubMed/MEDLINE, Web of Science, and Google Scholar. The comprehensive molecular biomarker database MarkerDB v2.0 (available at https://markerdb.ca/, accessed on 3 March 2025) was searched for experimentally verified biological markers in humans [62].

### 2.2. Search Strategy

The search string included free terms such as “sports nutrition”, “allostasis”, “body composition”, “allostatic load”, “allostatic overload”, and “physiological adaptation”. Each term was combined with keywords such as long-term, chronic, acute, nutrition, exercise, recovery, athletes, and biomarkers. The reference lists of the selected articles were also manually searched for additional literature (snowballing).

### 2.3. Findings Presentation

The authors collaborated remotely and contributed to the development of this article considering their standing and individual expertise in the field. Discrepancies were identified and resolved through discussion between authors where necessary. All communications and coordination throughout the process were completed electronically and were led by the first author. This article is organized into sections including (i) allostasis and adaptation; (ii) current approaches to biomarker monitoring in athletic populations; (iii) how to measure the allostatic load in athletes; and (iv) applied practice. Finally, future directions are presented to guide upcoming research in the field.

### 2.4. The 4Rs App

To ensure effective translation into practice, the “4Rs app” was developed within the free software environment for statistical computing and graphics R v4.4.0 [63] and the Shiny package v1.10.0 [64]. The development of the app had four phases: (i) conceptual design, where the educational goals of the 4Rs in sports nutrition were defined (including the allostatic load index and a scorecard for dietary supplements); (ii) UI/UX prototyping, where a responsive interface using HTML, CSS, and Shiny widgets was built; (iii) back-end integration, to connect interactive logic with nutrition algorithms via R scripts and reactive modules; and (iv) testing and deployment, where we tested the output, applied thematic styles (The 4Rs Framework branding), and deployed it via https://www.shinyapps.io/ (accessed on 11 April 2025). The app was registered in Zenodo (DOI: 10.5281/zenodo.15377939).

## 3. Allostasis and Adaptation

Stress is understood as the response to internal or external stimuli, referred to as stressors, which induce disruptions that exceed the usual physiological thresholds [65]. In simple terms, stress can be associated with a degree of perturbation within a biological system after exposure to a given stressor—this results in different stress types according to the type of perturbation, such as metabolic stress, psychological stress, mechanical stress, etc. Complementarily, allostasis refers to a biological system’s ability to adapt to daily challenges through predictive adjustments to maintain viability [66,67]. It is worth noting that allostasis and homeostasis are complementary components of integrative physiological regulation [68,69]. Allostasis is an extension of homeostatic parameters from single stable states to variational and relationally stable states—to clarify any doubts regarding the current conceptualization of homeostasis and allostasis, readers may refer to the article “Conceptual Foundations of Physiological Regulation Incorporating the Free Energy Principle and Self-Organized Criticality” by Bettinger and Friston (2023) [70].

Within the allostasis–interoception framework [71,72,73], and considering sports nutrition as a working example, chronic stimuli such as energy restriction, exercise-induced stress, or sleep disruption associated with pre-competition anxiety (external and internal stressors) can trigger systemic adaptations that recalibrate regulatory parameters to prioritize resource allocation towards activities that are critical for immediate survival (e.g., changes in substrate utilization, alterations in gene expression, exerkine regulations, changes in mood and focus, etc.). These factors—individually or, more likely, collectively—influence athletic performance by contributing to maladaptation.

As with any biological system, the human adaptation process needs two critical factors: energy and time. The energy factor has recently been defined as “allostasis and stress-induced energy expenditure” by Bobba-Alves et al. (2022) [74]. In athletes, this consequently influences the total daily energy expenditure, as has been discussed for injured athletes [58,75]. This approach may also provide insights into individual responses during nutrition and exercise interventions, potentially explaining phenomena at different organization levels, such as leptin pathway alterations, reductions in resting energy expenditure, menstrual cycle irregularities, and increased susceptibility to lean mass loss—all hallmarks of energy restriction-induced stress [76,77]. The time factor is connected to the restorative period required to repair tissue and to correct interoceptive prediction mistakes, as well as the many mechanisms that keep internal conditions within the new setpoint of adaptation. For example, this includes recovery periods between exercise sessions and adequate sleep for psycho- and physiological recuperation [1] (Figure 2). As a result, both energy and time factors influence each individual/population’s allostatic load, which can be described as the cost that a biological system must pay in order to reset its physiological parameters during the adaptation process [78].

It is worth noting that Chrousos (2009) introduced the term cacostatis (“bad state”, from Ancient Greek κακός [*kakós*], which means “bad”) to refer to this state of disharmony and the cacostatic load as the cumulative pathophysiological burden of the organism [79]. Following this etymological conceptualization, the beneficial adaptation that leads to the improved capacity of the human body might be referred to as hyperstasis (“higher better state”) [80].

The allostatic load can increase dramatically if the system has superimposed on it additional chronic loads (inactivity or overactivity) that exceed the capacity to cope (e.g., psychological disorders, cancer, under-nutrition, or drug-induced side effects [including steroids]), in so-called allostatic overload [81]. Allostatic overload has been defined as “long-term energy-dependent functional and/or structural dysregulation and breakdown that arise due to chronic allostatic load, leading to accelerated aging, disease onset and progression, and increased mortality risk” [74,82] or cacostasis [80]. In the context of this work, physical exercise-induced stress relies on individual responses (prior knowledge plus each R of our 4Rs operational framework) that follow an inverted U-shaped allostatic response (Figure 3).

Thus, under the allostasis–interoception paradigm, we can consider syndromes such as overtraining, burnout, or relative energy deficiency in sport (RED-S) as cacostasis or manifestations of allostatic overload in athletes. The difference between these syndromes depends on the clinical variables that are used to monitor the athlete and the organic system (physiological context) that is being evaluated. Overall, these clinical manifestations appear when the athlete exceeds their capacity to cope with exposure to chronic stress (e.g., nutrition and exercise-caused low energy availability or sleep disturbances). We should remember that any biological adaptation process critically needs energy and time.

Understanding recovery after physical effort as a dynamic process involving continuous adjustments/resets of biological parameters underscores the significance of the quality, magnitude, duration, timing, and novelty of the stressor, alongside the individual pre-existing state information of the athlete (e.g., training history, current body composition, genotype, etc.). Our group has emphasized that adopting this systemic (integrative and multifactorial), evolutionary (biologically adaptive by nature), and adaptive (dynamic and individualized) perspective, or “bio-logic approach” [2,83,84], would enhance our understanding of the flow of information through interactions between system components and their regulatory aspects for a given phenotype and its allostatic load (Figure 4). Importantly, nutrition is one of the many factors that might impact the allostatic load and, thereby, it might influence musculoskeletal tissue overload and repair [58].

## 4. Current Approaches to Biomarker Monitoring in Athletic Populations

Biomarker profiling is applied to the monitoring of health, performance, and recovery. Existing approaches to assessing physiological states, nutrition optimization, and post-training/competition recovery all utilize blood biomarkers. Principal among these are the biomarkers related to inflammatory cytokines, muscle enzymes, and metabolic markers, which show a response in the athlete due to exercise and fatigue resulting from the adaptation to it [87]. However, their usefulness as biomarkers is usually compromised by several factors: individual variability, the need for the standardization of the measurement methodologies, and the complexity of result interpretation in the context of multivariate training environments [88,89].

Lee and colleagues (2017) [90] suggested a comprehensive approach to biomarker analysis with categories that enable coaches and athletes to monitor performance, recovery, and health in an individualized way. By using several validated biomarkers, the sensitivity is increased, allowing for the early detection of training, recovery, or diet impacts. Long-term biomarker data can help to prevent injuries and performance declines, with examples of well-studied biomarkers suggested for the creation of a customized monitoring panel (Figure 5).

Reliable biomarker technologies and individualized monitoring approaches are needed to further enhance the accuracy of current tools for high-performance cohorts. We recently proposed a human–machine collaboration method to identify potential biomarkers of exercise stress-induced immunodepression linked to iron metabolism using network topology analysis and bioinformatics [83]. In this regard, Figure 6 shows the response pattern of the immune function to exercise-induced stress, although individual variation (i.e., prior knowledge) should be considered, and it should be recognized that several other factors (e.g., psychological stress, under-nutrition, poor sleep, and long-distance travel) can also lower immunity in athletes [91]. Recently, Haller et al. (2023) described recommendations for best practices in the use and interpretation of biomarkers to support workload management in sport-specific contexts [92]. Hence, diagnosis and monitoring encompass the screening of an athlete’s health while monitoring the load and recovery. The authors suggested practical biomarkers with reference ranges, interpretations, and correlations with the training load, as well as questionnaire scores that are categorized into systems such as muscle, immune, metabolic, and inflammation.

### 4.1. Clinical Assessment Tool Version 2 of the International Olympic Committee

As discussed earlier in this article and in previous works [1,58,83,84], relative energy deficiency in sport (RED-S) may be considered a manifestation of allostatic overload or cacostasis in athletes. The working group of the International Olympic Committee (IOC) defines RED-S as a syndrome of impaired physiological and/or psychological functioning experienced by female and male athletes, caused by exposure to problematic (prolonged and/or severe) low energy availability [93]. This might be seen as a disequilibrium state where the individual’s dietary energy intake is insufficient to support the energy expenditure required for health, function, and daily living once the cost of exercise and sporting activities is considered.

The IOC recently introduced the Clinical Assessment Tool Version 2 (CAT2) to evaluate athletes and active individuals suspected of experiencing problematic low energy availability resulting in RED-S. The tool also provides guidance on determining appropriate levels of sport participation [94]. It is intended for use by athlete health and performance teams, under the guidance of a physician, for the clinical assessment and management of athletes with RED-S. The CAT2 consists of a three-step operational framework: Step 1—RED-S screening using population-specific questionnaires assessing the presence of RED-S indicators or clinical interviewing; Step 2—RED-S severity/risk assessment and stratification and sports participation guidelines; and Step 3—RED-S clinical diagnosis and treatment, which uses information collected in the previous steps. While the CAT2 tool offers several primary and secondary indicators (used in Step 2) and proves practical for use with elite athletes, the authors note that some potential indicators with subclinical or clinical deviations are intentionally left vague in their quantification. This approach reflects the need for further research to refine the parameters and establish more precise cut-offs. Future updates to the tool are anticipated to address sport-specific populations, accounting for the many confounding factors that must be considered when interpreting individual athlete profiles [89].

Jeukendrup et al. (2024) introduced an Athlete Health and Readiness Checklist (AHaRC) that compiles tools previously developed and validated by various expert/consensus statements to monitor and troubleshoot aspects of athlete health and performance issues (Figure 7) [95]. Although our 4Rs framework of sports nutrition and its foundations in the allostatic load in the sports context (Figure 3) were introduced in 2020 [1], we agree with Jeukendrup and colleagues that most practitioners (exercise and nutrition professionals) and sports scientists largely ignore the concept of allostasis. As we will discuss in more detail in the following sections, we propose that gaining a systemic and comprehensive understanding of the nutritional and physical exercise-induced cumulative physiological burden that athletes are experiencing—which is essential for diagnosis and risk stratification—requires an evaluation of several components of allostatic load, including cardiovascular, metabolic, immune, and neuroendocrine factors.

### 4.2. The Reliance on Body Composition to Estimate Energy Availability

Energy availability (EA) refers to the energy remaining for metabolic functions after subtracting the exercise energy expenditure (EEE) from energy intake (EI) [96]. It is calculated as EA = (EI [kcal] − EEE [kcal])/FFM (kg), where FFM is fat-free mass [93]. To support healthy physiological function in athletes, EA should be ≈45 kcal/kg FFM/day, while levels below 30 kcal/kg FFM/day indicate low energy availability and should be avoided [10,58,75,93,96]. In reality, this strict definition of LEA is somewhat impractical in the field because it requires very accurate measures of the dietary EI, EEE, and body composition. The EEE is mostly influenced by the actual training load, which may differ from day to day. Accurate measurements of EI and FFM are also difficult to achieve in the field, so the need for intervention is usually decided on the basis of identifying symptoms and measurements of biomarkers (e.g., hematological parameters and selected plasma hormones known to change with LEA).

In this context, changes in body composition are the outcome of physiological adaptations to chronic intermittent stressors that unfortunately cannot be evaluated with a single technique due to the drawbacks of all body composition methods. In addition, successful exercise and nutrition intervention relies on the continuous and reliable assessment of body composition. The four-compartment model represents the current gold-standard approach and combines different techniques to measure/estimate the components of fat mass, total body water, bone mineral content, and residual content. Unfortunately, this is limited to research and diagnostic laboratories. In sports practice, a relatively accessible method is dual-energy X-ray absorptiometry (DXA). However, practitioners should be aware that the FFM estimated with DXA should be corrected for the fat-free adipose tissue (FFAT) as this can change the interpretation of the results during a dietary/exercise intervention [97] or during sarcopenia diagnosis [98,99]. Practitioners and sports scientists are advised to use the term FFM instead of lean body mass, as FFM aligns with chemically accurate terminology. For a recent critical review on this topic, refer to Heymsfield et al. (2024) [100]. In agreement with the authors, adopting the correct terminology will help to minimize confusion and enhance scientific rigor in body composition research.

If DXA is inaccessible, practitioners frequently use other in-field and more affordable techniques, such as bioelectrical impedance analysis (BIA) and kinanthropometry. However, these methods may present higher variability when the international recommendations of the International Society for Electrical Bioimpedance (ISEBI) or the International Society for the Advancement of Kinanthropometry (ISAK) are not followed. It should be noted that there is a high prevalence of poor analyses based on non-specific-population equations; therefore, when monitoring body composition, we strongly recommend that clinicians and practitioners focus their analysis on the absolute values of these techniques, such as the outcomes of the bioelectrical impedance vectorial analysis (BIVA; phase angle and Xc/R graph) for BIA [101] or the sum of skinfolds and skinfold-corrected girths as adiposity and muscularity indices, respectively, for kinanthropometry [102] (Table 2). Readers may refer to the review article by Barakat et al. (2020) [103], which describes the advantages and disadvantages of different techniques for the estimation of body composition, as well as to Beausejour et al. (2024) for the assessment of skeletal muscle health [104].

## 5. How to Measure the Allostatic Load in Athletes?

First of all, it is important to recognize that the allostatic load represents the cost that a biological system (e.g., an individual organism, tissue, or cell) incurs to achieve a new (allostatic) state of adaptation [1,74]. As we highlighted in our first article on the 4Rs framework of sports nutrition [1], to effectively structure a recovery process, it is essential to prepare the body both physically and mentally to meet the demands of training and competition, while ensuring that the adaptation processes necessary for progressive improvement are not disrupted [105]. As a result, recovery strategies will largely depend on factors such as the timing of the next session, the level of physiological stress, and the importance of the upcoming event. These factors guide the approach to rehydration, energy replenishment, and nutrient intake required for optimal tissue repair and recuperation. Thus, the correct use of the 4Rs framework—Rehydration, Refuel, Repair, and Recuperate—would reduce the components of the allostatic load and ultimately increase the effectiveness of the intervention, allowing the athlete to reach their goal with less effort and in less time [1].

To measure the cumulative physiological burden on the body due to chronic stress, clinicians and researchers use the allostatic load index (AL_index_). This index typically includes several biomarkers that show either sub-clinical or clinical deviations to assess the overall strain on the main physiological systems: cardiovascular, metabolic, immune, and neuroendocrine [74]. Generally, these biomarkers are equally weighted and summed to calculate an index ranging from 0 to approximately 9–12. Figure 8 displays the frequency of biomarker use in the allostatic load index in neighborhood research. Biomarkers were extracted from 18 articles identified through a systematic review [106].

Table 3 shows the original set used in clinical practice and a proposed updated version of the biomarkers used to calculate the AL_index_, with the corresponding cut-off points in chronic disease populations, based on available evidence [107,108,109]. Although non-communicable diseases, such as diabetes, cancer, and heart disease, are beyond the scope of this article, readers are encouraged to increase their awareness of their importance and use “the healthy trinity” of physical activity, nutrition, and sleep as a shield for the prevention and management of chronic diseases, as well as using the allostatic load index for clinical monitoring. Please refer to Bonilla et al. (2024) [110] for the complete description of this operational framework, which contributes to Sustainable Development Goal 3.

The selection and threshold determination of these biomarkers remain a prominent focus in both basic and applied research on health and disease across different populations [108,109,111,112,113,114,115,116]. Conversely, although the foundational contributions by Sterling and Eyer (1988) [67], McEwen and Stellar (1993) [117], and McEwen [78] emphasized that allostasis involves the entire brain and body, rather than localized feedback mechanisms—and highlighted exercise as an example of a stressor and physical condition as a modulator of individual responses to stressors—significant work remains in the field of exercise and sports sciences to develop a robust AL_index_ for athletes. In the 2000s, several researchers contributed regarding the regulatory and protective role of physical exercise in the allostatic load [118,119], while others suggested physiological mechanisms by which physical activity may improve physical fitness through allostasis [120]. Rabey and Moloney (2022) recommended allostasis as a possible explanatory model for the onset and maintenance of pain, which can integrate the allostatic load into clinical reasoning to guide decision-making [121]. We also adopted an allostatic approach to explore the etiology of pain and common injuries among fitness exercisers [2]. Interestingly, the association between allostatic load and overuse musculoskeletal injury in the US Marine Corps was confirmed recently by Feigel et al. (2024) [122]. Moreover, Forys and Tokuhama-Espinosa (2022) analyzed the relation between neurotransmitters and allostatic load in an effort to understand the occurrence of depression in elite athletes, suggesting that exercise-induced neurohormonal imbalances might contribute to depressive symptoms [123]. In 2023, we also argued that the individual effects of physical activity and certain adaptogens on depression-related outcomes could be explained by reductions in allostatic load, providing a rationale for synergism [124].

It is essential to point out that current conceptualizations describe interoception as a model and allostasis as control [71], both operating under the free energy principle proposed by Dr. Karl Friston [125]. Under this principle, Inui (2023) argued that allostasis not only requires the integration of interoceptive but also exteroceptive and proprioceptive signals [126]. Allostasis can be seen as a mechanism with which the body adjusts its physiological parameters within a range that is harmless (e.g., increasing oxygen delivery to muscles during intense exercise, lowering heart rate during post-workout recovery, and redirecting blood flow from the digestive system to active muscles during physical activity), in response to changes anticipated from past experience, aimed at minimizing the future surprise (i.e., free energy). In this context of mathematical analysis and dynamical systems that follows Bayesian inference, allostasis is currently referred to as “variational and relational stability” by Bettinger and Friston (2023), which expands upon the traditional concept of “stability through change” and highlights the inherent capacity of organic systems for physiological resilience, or the ability to “return to stability”. These authors also describe that allostatic health encompasses methods associated with psychoneuroimmunology [70]. Similarly, Tossici et al. (2024) suggest using allostatic load to monitor psycho-physical condition in sports performance as a psychoneuroimmunology-inspired approach [127].

Very few studies have directly assessed biomarkers to include in the AL_index_ for the athletic population. In a pilot study, we recently evaluated the relationship between several molecular, physical, and psychometric biomarkers with the goal of contributing to the development of a valid AL_index_ for Colombian professional basketball players (*n* = 12). Interestingly, the levels of creatine kinase (CK) and the session rating of perceived exertion (also known as Foster’s index) showed significant correlations with an arbitrary cumulative score that reflected the exercise-induced cumulative physiological burden [128]. While we did not evaluate hormonal markers, Milheiro et al. (2024) found a relationship between saliva cortisol, insulin, and rating of perceived exertion responses in nine ultrarunners (average time to completion of the ultramarathon = 24 h) [129], as part of a doctoral thesis titled “Impact of allostatic biomarkers on health and performance”. Recently, Feigel et al. (2025) found that an elevated allostatic load (using an eight-item AL_index_) was significantly associated with physical and psychological maladaptation in military personnel (*n* = 31, 14F) undergoing a 10-week military physical and tactical training course [130].

Based on the comprehensive panel and clinical evidence of well-studied biomarkers in sports, Table 4 presents a proposal for the calculation of the AL_index_ for physique practitioners (whether they use performance and image-enhancing drugs [PIEDs] or not) [131,132,133,134,135,136] and elite athletic populations [87,88,90,92,137,138,139,140,141,142]. To calculate the AL_index_, each biomarker is equally weighted and summed, resulting in an index ranging from 0 to 10. A value of 1 is assigned if the biomarker exceeds the cut-off point, while a value of 0 is assigned if it falls within the normal range. This clinical risk score has been utilized in the allostatic load literature and refined with population- and sex-specific features derived from available clinical evidence in exercise and sports sciences. It is important to note that the AL_index_ is highly versatile, as the choice of scoring algorithm for biomarkers (i.e., high-risk quartiles, two-tailed high-risk quartiles, high-risk deciles, two-tailed high-risk deciles, Z-scores, system weighted, sex-specific cut-points, or clinical) does not significantly impact the prediction of health-related outcomes. This remains valid provided that sex-specific cut-offs are applied when appropriate, at least in the general population, as evidenced by McLoughlin et al. (2020) [143].

In an athletic population, all blood markers should be assessed in a rested and fasted condition. The required evaluation time varies depending on the biomarker, as each has a different response and clearance rate following exercise. For example, (i) the resting blood lactate concentration (b[La^−^]) should be assessed within 12–24 h post-exercise, with higher values exceeding 1.5–4 mmol/L; (ii) the free testosterone-to-cortisol ratio (fT/C), S100 calcium-binding protein B (S100B), and hepcidin require at least 24 h; and (iii) CK levels peak between 48 and 72 h (considering inter-athlete variability, as elevated CK typically ranges between 200 and 1000 U/L) [144]. Therefore, there is potential flexibility to adapt biomarkers in a sport-specific manner or apply alternative cut-off points beyond those proposed in this article (Table 4), provided that the clinical information is based on high-quality scientific evidence, such as an umbrella review of systematic reviews and meta-analyses or a Delphi-based international consensus. 

**Table 4 life-15-00867-t004:** Proposed allostatic load indices for athletic populations.

	Biomarker	Cut-Off Point	System	MarkerDB	Clinical Evidence	References
AL_index_ in Physique Athletes	fT/C (natural athletes) *	Reduced by +30%	Neuroendocrine	No	★★★★☆	[134,145]
	HDL cholesterol	<40 mg/dL (M); <50 mg/dL (F)	Cardiovascular	Yes	★★★★★	[131,146,147,148]
	Aspartate transaminase	>40 U/L	Metabolic	Yes	★★★★★	[131,135,146,147]
	Alanine aminotransferase	>40 U/L	Metabolic	Yes	★★★★☆	[131,135,147]
	Cystatin C	>1.0 mg/L	Metabolic	Yes	★★★☆☆	[149,150]
	C-reactive protein	>0.3 mg/dL	Inflammatory	Yes	★★★☆☆	[90]
	Session RPE **	Increased	Neuroendocrine, Cardiovascular	NA	★★★★☆	[136,151]
	Cardiac troponin T	>0.01 ng/mL	Cardiovascular	Yes	★★★★☆	[152,153]
	Creatine kinase	>200 U/L (M); >170 U/L (F)	Inflammatory	Yes	★★★★☆	[132,138]
	Blood pressure instability	≥140 mmHg (SBP)	Cardiovascular	Yes	★★★★☆	[133,147,154,155]
AL_index_ in Elite Athletes	fT/C ***	Reduced by +30%	Neuroendocrine	No	★★★★☆	[156,157]
	Heart rate variability	Instability (individual basis)	Neuroendocrine, Cardiovascular	Yes	★★★★☆	[158,159,160]
	S100B	>0.12 µg/L	Neuroendocrine	Yes	★★★★☆	[161,162,163]
	Hepcidin	Increased	Inflammatory	Yes	★★★★☆	[164,165]
	Energy availability	<30 kcal/kg FFM/day	Metabolic	NA	★★★★☆	[166,167]
	C-reactive protein	>0.3 mg/dL	Inflammatory	Yes	★★★☆☆	[88,90]
	Session RPE **	Increased	Neuroendocrine, Cardiovascular	NA	★★★★☆	[168,169]
	Resting b[La^−^]	Increased	Metabolic	Yes	★★★★☆	[141,142,156]
	Creatine kinase	>200 U/L (M); >170 U/L (F)	Inflammatory	Yes	★★★★☆	[138,144,156]
	25-hydroxy vitamin D	<20 ng/mL (deficiency)	Neuroendocrine, Metabolic	Yes	★★★★☆	[170,171]

A new proposal to evaluate allostatic load in athletic populations. Practitioners should note that further research is still needed to validate these sets of biomarkers in exercise and sports settings across different disciplines, populations, sexes, and ages. MarkerDB v2.0 is a freely available electronic database that attempts to consolidate information on all known clinical, and a selected set of pre-clinical, biomarkers into a single resource. NA: Not applicable—refers to composite or derived biomarkers (e.g., session RPE and energy availability) that are not included in MarkerDB v2.0, as they are calculated using several variables rather than being individual molecular markers. Clinical evidence for each biomarker is categorized as follows: ★★★★★ for either umbrella reviews of systematic reviews and meta-analyses or Delphi-based international consensus; ★★★★☆ for systematic reviews with meta-analyses; ★★★☆☆ for systematic reviews. b[La^−^]: blood lactate concentration; F: female; fT/C: free testosterone to cortisol ratio; HDL: high-density lipoprotein; M: male; MA: meta-analysis; RPE: rating of perceived exertion; S100B: S100 calcium-binding protein B; SBP: systolic blood pressure; SR: systematic review. * Hair cortisol might be used for PIED users (>150 pg/mg); ** session RPE = RPE x session duration (min), also known as Foster’s index; *** at least in men, the testosterone–estradiol ratio is reduced by ~50% in overtrained athletes compared with healthy athletes; therefore, when compared to fT/C, it has been suggested as a better predictor of overtraining [172].

## 6. Applied Practice

To effectively apply this novel AL_index_ in athletes, several questions might arise: How frequently should biomarkers be measured? How should borderline biomarker levels be interpreted? How can nutrition or exercise training be adjusted based on AL_index_ data? Although further research is needed to answer these questions accurately, the following section describes how a daily and weekly AL_index_ could guide immediate adjustments to the training load or nutrition prescriptions, ultimately translating conceptual insights into practical applications.

### 6.1. Training Feedback

Coaches and strength and conditioning staff should meet weekly to review AL_index_ trends alongside performance metrics such as the shooting accuracy, sprint times, and jump testing, among others. As the AL_index_ is reduced and on-court indicators improve, the training volume can gradually be increased to ensure continued progress without risking overtraining or injury. If the AL_index_ remains elevated or if performance data decline, the team can adjust the training load (e.g., reducing plyometric work) or reinforce the recovery strategies of the 4Rs, such as pre-sleep protein intake and hydration protocols. In another hypothetical situation, if a subset of athletes shows consistently rising CRP and CK, coupled with a higher session RPE, coaches could reduce the upcoming week’s high-intensity interval training sessions or schedule an extra day of active recovery. This adjustment could help to mitigate further strain on the athlete’s body and promote recovery. Similarly, a drop in HRV readings below an individual’s baseline may indicate increasing cumulative stress. In such cases, a brief taper (e.g., reducing the total training volume by ~20%) could help to restore neuromuscular readiness before the next high-intensity training session.

### 6.2. Adjusting Nutrition

Consider a collegiate basketball team collecting weekly AL_index_ data throughout their pre-season conditioning (e.g., pre-season training camp). If resources are available, the AL_index_ could include biomarkers such as resting b[La^−^], HRV, CRP, CK, fT/C (or testosterone–estradiol ratio), alongside self-reported fatigue levels (e.g., session RPE). Athletes whose AL_index_ scores remain high for two consecutive weeks might be advised to increase their carbohydrate intake, especially around key training sessions. Increasing carbohydrate availability from 5–6 g/kg/day to 6–8 g/kg/day could enhance muscle glycogen restoration and reduce metabolic stress, aiding in recovery and performance. Similarly, in cases where high markers of muscle soreness or damage persist, the sports dietitian could recommend more frequent protein feedings (e.g., 4–5 doses daily) or slightly higher doses (e.g., 0.4 g/kg per meal) to support muscle repair. Adding 25–30 g of protein before sleep might further enhance overnight recovery, helping the athlete to repair damaged tissue and optimize muscle growth.

As previously demonstrated, biomarker monitoring can enhance positive adaptations and reduce injuries caused by stressors during a competitive season [173]. However, collecting between 60 and more than 92 biomarkers during pre-season and in-season weeks presents a significant challenge for coaches and nutritional/medical staff. By applying the AL_index_ to adjust training loads, nutrition, and recovery strategies based on real-time physiological data, coaches, dietitians, and athletes can optimize performance and reduce the risk of overtraining and injury while assessing fewer biomarkers. Notably, when using the AL_index_ to monitor athletes, there is flexibility to build a sport-specific index of up to 10 biomarkers (or fewer) or use different cut-off points beyond those presented in this article, primarily relying on clinical evidence. It is essential to consider the 4R framework of sports nutrition to optimize post-exercise recovery and support the athlete’s allodynamic response. These four factors can help guide nutrition and recovery strategies while mitigating allostatic overload (Figure 9).

To ensure effective translation into practice, the 4Rs app was developed (available at https://dbss.shinyapps.io/4RsApp/, accessed on 22 April 2025). This application would help performance and health nutrition professionals in designing personalized recovery and nutrition plans through a simple-to-use interface. It includes tools to calculate a 4R recovery protocol and score the AL_index_ using biomarkers with abnormal values (see Table 3 and Table 4) and provides evidence-based recommendations for the selection of a dietary supplement by prioritizing quality and safety (Figure 10). Considering the inter-individual variability among athletes and active individuals, the application includes different sports and activities (e.g., powerlifting/weightlifting, bodybuilding, endurance, sprints/team sports, among others), as well as exercise intensities and conditions (e.g., moderate duration/low-to-moderate intensity, maximal intensity/competition, bulking, cutting, injury, etc.). Furthermore, the 4Rs app contains educational content (e.g., videos, figures), key references, and frequently asked questions about the allostatic load to enhance scientific literacy among medical staff, nutritionists, coaches, and athletes. Health and sports practitioners may consider the AL_index_ as a diagnostic tool for maladaptation, while the 4Rs of sports nutrition provide an important pillar of individualized intervention for recovery and resilience.

## 7. Limitations and Future Directions

High-level sports already assess a wide array of biomarkers in athletes, along with movement and sleep data, psychological measures, and practice, training, and performance analytics, to monitor them. This is done to evaluate training progress, support recovery, optimize performance, and prevent overtraining or RED-S. The proposed AL_index_ offers a way to harness and interpret all of these data for more effective athlete management. There is no doubt that the proposed set of biomarkers requires further research to confirm their validity and generate more accurate clinical recommendations for specific sports, sexes, ages, and training phases. It needs to be noted that the available evidence indicates that neither the menstrual cycle [174,175,176] nor oral contraceptive use [177,178] significantly impacts exercise adaptations or physical performance in women; however, individual monitoring and sex-specific cut-offs may be warranted for athletes prioritizing elite performance.

The current information is primarily based on associations, underscoring the need for more comprehensive studies that explore the longitudinal interactions between allostatic overload (cacostasis) and sports performance. Additionally, as with other studies in the field, there is a significant gap in terms of multilevel analyses, causal modeling, and complexity-based approaches in the existing research [73]. Finally, more sensitive laboratory methods are required to be included in sports practice to improve diagnosis and reliability [179].

We encourage exercise scientists and sports nutrition researchers to contribute to the development and validation of the AL_index_ for athletes as a multisystemic clinical tool to evaluate and monitor the adaptation process. This will help to refine and reduce the large number of biomarkers (over 60) proposed in recent years (Figure 5) [90], as well as to avoid oversimplification and inaccurate reliance on a few biomarkers in isolation, which can lead to misdiagnosis. It is worth noting that, based on recent multi-cohort meta-analytic findings in mid-to-late-life populations (*n* = 67,126 participants aged 40 to 111 years, ~50% women) reported by McCrory et al. (2023) [107], a five-item AL_index_ could also be developed using fewer biomarkers when resources are limited. Similarly, Polick and colleagues (2024) [112] emphasized the value of a six-item AL_index_ using multi-wave analysis as a predictor of morbidity and mortality, particularly in older adults, and demonstrated that an individualized AL_index_ aligns with the shift towards precision medicine. From bench (lab) to bedside, this is particularly important for PIEDs abuse in recreational exercisers and amateur bodybuilders, which is increasingly recognized as a serious public health concern [180,181,182]. Future research should focus on the versatility of the AL_index_, as the choice of scoring algorithm for biomarkers does not significantly affect the prediction of health-related outcomes, as long as they use sex-specific cut-offs when needed [143]. Personalized ranges based on an athlete’s history may provide a more robust version that could enhance the performance of the AL_index_. In this regard, Hecksteden et al. (2017) demonstrated that developing individualized reference ranges using a Bayesian approach enhances accuracy in athlete recovery monitoring, accounting for interindividual variability [144].

In agreement with Mass et al. (2015) [183] and Pedlar et al. (2019) [88], key components for biomarker suitability that should be considered by practitioners and researchers include the following.

Evidence: Is there sufficient research supporting the biomarker’s use in the target sport population and sex? The main features of the population (e.g., age, sex, sport setting, place of recruitment) should be described.Application: Does the biomarker provide actionable data, serving as a reliable indicator for positive or negative outcomes? Are cut-off values presented for each outcome variable (neuroendocrine, inflammatory, cardiovascular, and metabolic) used in calculating the AL_index_?Validity: Has the biomarker been validated, especially if it is a new technique, and does it align with established “gold standard” methods?Variability: Is the biomarker’s analytical and biological variability acceptable, often measured as the coefficient of variation?Collection and analysis: Is the biomarker collection and analysis procedure fast and efficient, with minimal blood required?Sample treatment and transportation: Can the sample be analyzed on-site, or does it require specific storage and transportation conditions?Diurnal variation: Are there factors, such as the time of day, exercise, sleep, and fasting status, that influence the biomarker?Cost: Is the cost of obtaining biomarker data justified by its utility?Covariates: Are there known factors, such as environmental conditions, altitude, or travel stress, that influence the biomarker’s reliability?Data analysis: Have the analyses been adjusted for potential sources of confounding? Is the number of cases in the multivariate analysis at least 10 times the number of independent variables? Present measures of associations (e.g., odds ratios including 95% confidence intervals for logistic regression, β for linear regression). The AL_index_ should be calculated using the standardized method of risk quartiles and/or clinical sex-specific cut-offs.

## 8. Conclusions

The 4Rs of sports nutrition (Rehydrate, Refuel, Repair, and Recuperate) represent an evidence-based operational framework designed to enhance post-exercise recovery, grounded in the allostasis paradigm. Under the allostasis–interoception paradigm, we can consider syndromes such as overtraining, burnout, or RED-S as cacostasis or manifestations of allostatic overload in athletes. Despite its accuracy and robustness in clinical settings for the assessment of morbidity and mortality, the allostatic load index (AL_index_) remains an underutilized tool that could effectively assess the exercise-induced cumulative physiological burden in athletes. In this article, we cautiously propose an AL_index_ based on 10 biomarkers tailored to physique competitors (with or without the use of PIEDs) and elite athletes. To calculate the AL_index_, each biomarker is equally weighted and summed, resulting in an index ranging from 0 to 10 (a value of 1 is assigned if the biomarker exceeds the cut-off point, while a value of 0 is assigned if it falls within the normal range). The timing of blood draws is crucial for accurate readings—samples should be collected in a rested and fasted state and not immediately after a high-intensity workout. Importantly, practitioners should be cautious not to confuse system complexity with clinical difficulty or impracticality. For instance, recent findings demonstrate that even a simplified, five-item AL_index_ has been validated to effectively capture physiological “wear and tear” and predict mortality in the general population [107]. Thus, practitioners can use the AL_index_ as a diagnostic tool for maladaptation, with the 4Rs framework then providing the essential structure for customized recovery and resilience strategies.

We invite practitioners and researchers to adopt a systemic approach that uses the available athlete data, emphasizing a predictive, integrative, and allostatic perspective. This approach considers the dynamic and individualized adaptive responses—referred to as “bio-logic” [2,83,84,110]—to more accurately capture the inherent complexity of human physiological resilience, as conceptualized by Bettinger and Friston (2023) [70]. Thanks to the versatility of the AL_index_, several indices are expected to be developed and validated for different exercise and sports populations. Personalized ranges based on an athlete’s history may provide a more robust version that could enhance the performance of the AL_index_. We acknowledge that our proposal of the AL_index_ in athletic populations is limited to evidence from its individual components, and further research is necessary to validate the full set of biomarkers and AL_index_ during longitudinal or interventional studies. Thus, we encourage exercise scientists and sports nutrition researchers to contribute to the construction and validation of this multisystem assessment measure, which aims to monitor the adaptation processes in athletes and physically active individuals.

## Figures and Tables

**Figure 1 life-15-00867-f001:**
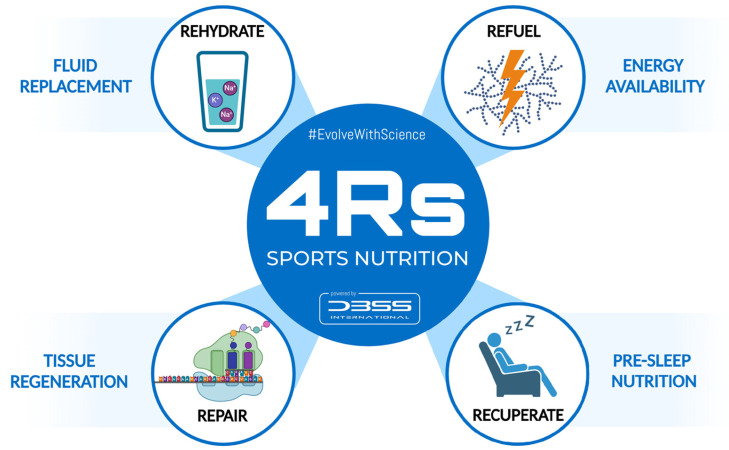
The 4Rs framework of sports nutrition. Source: designed by the authors (D.A.B.).

**Figure 2 life-15-00867-f002:**
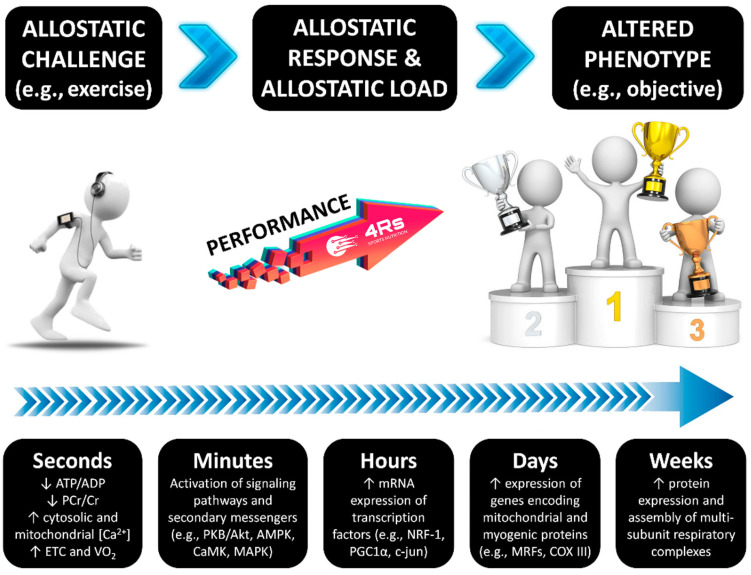
The time course of exercise-induced adaptation. Transient changes in metabolite sensing and signaling during/after exercise drive the gene transcription of early genes; myogenic regulators; genes of carbohydrate metabolism, lipid mobilization, transport and oxidation, mitochondrial metabolism, and oxidative phosphorylation; and transcriptional regulators of gene expression and mitochondrial biogenesis. Source: designed by the authors (D.A.B.). Refer to Bonilla et al. (2020) [1] for further information.

**Figure 3 life-15-00867-f003:**
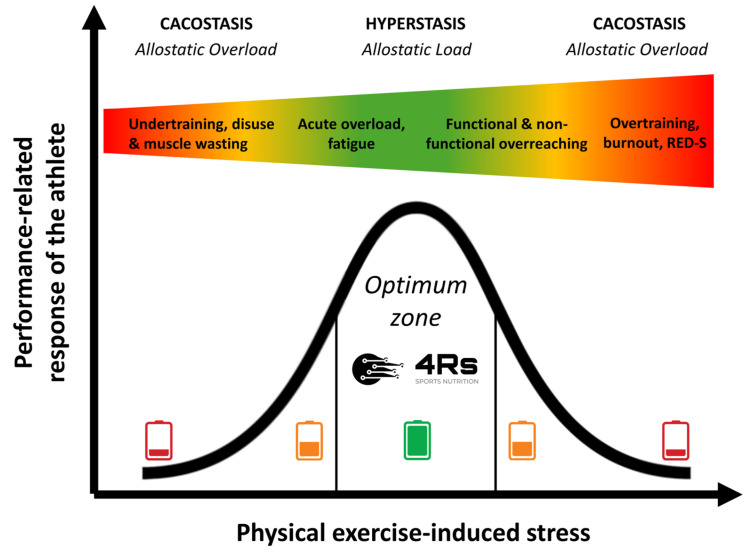
Performance-related response of an athlete to chronic physical exercise-induced stress stimuli over time. The color gradient visually represents the transition from optimal adaptation (green) to excessive stress (red). Source: designed by the authors (D.A.B.).

**Figure 4 life-15-00867-f004:**
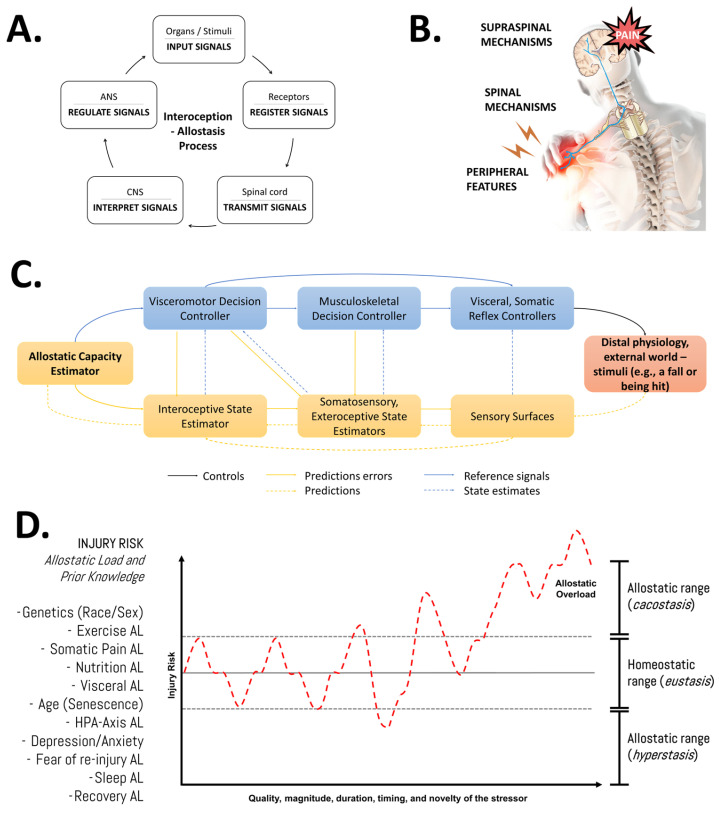
An integrative view of the multifactorial nature of the injury risk in athletes. (**A**) General features of the allostasis–interoception process. While allostasis represents the adaptive process of stability through change, interoception refers to encoding representations of the internal (physiological) state of the body [85]. (**B**) Modulation of endogenous pain. Nociplastic pain conditions include the combination of central and peripheral pain sensitization, hyper-responsiveness to painful and non-painful sensory stimuli, and associated features (fatigue, sleep, and cognitive disturbances) [86]. (**C**) Detailed representation of the allostatic–interoceptive control (as a closed-loop system) of the human body in response to any stimuli. The injury or pain etiology might be discussed in terms of the role of the input signal (stimuli—distal physiology or external world), receptors (sensory surfaces, biological receptors), transmitters (spinal cord, anatomy trains), decoders (central nervous system), regulator elements (autonomic nervous system), and output signal (response, physiological effects). This block diagram was taken from Sennesh et al. (2022) [71]. (**D**) Schematic illustration of the allodynamic response to stress in sports injuries. AL: allostatic load; HPA: hypothalamic–pituitary–adrenal axis. Source: designed by the authors (D.A.B.). Refer to Bonilla et al. (2022) [2].

**Figure 5 life-15-00867-f005:**
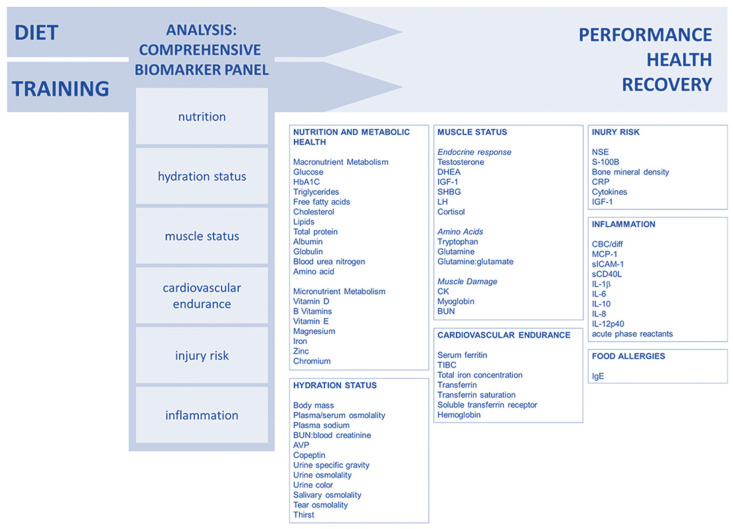
Comprehensive approach to biomarker analysis. Reproduced from Lee et al. (2017) [90]—Creative Commons Attribution 4.0 International License.

**Figure 6 life-15-00867-f006:**
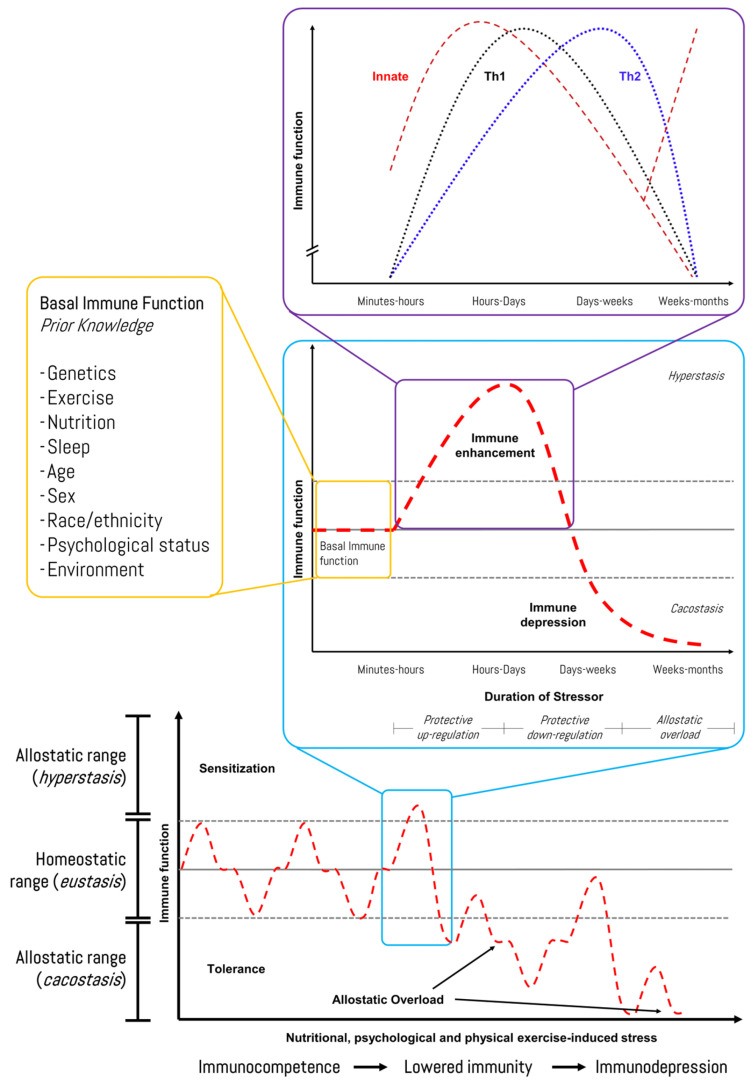
Immune-related response of an athlete during a time course of chronic stress stimuli in sports. Source: designed by the authors (D.A.B.). Refer to Bonilla et al. (2022) [83].

**Figure 7 life-15-00867-f007:**
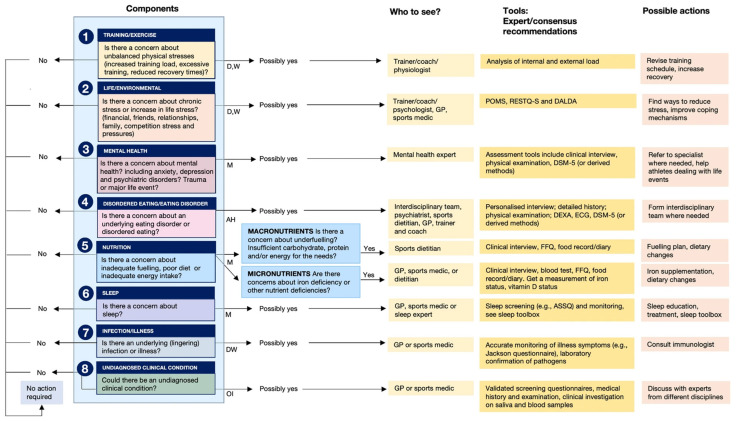
Athlete Health and Readiness Checklist (AHaRC), providing a multidimensional decision tree to maintain athletes’ health and performance. AH: ad hoc; D: daily; M: monthly; OI: on indication; W: weekly. Reproduced from Jeukendrup et al. (2024) [95]; refer to this for more details related to the AHaRC tool—Creative Commons Attribution 4.0 International License.

**Figure 8 life-15-00867-f008:**
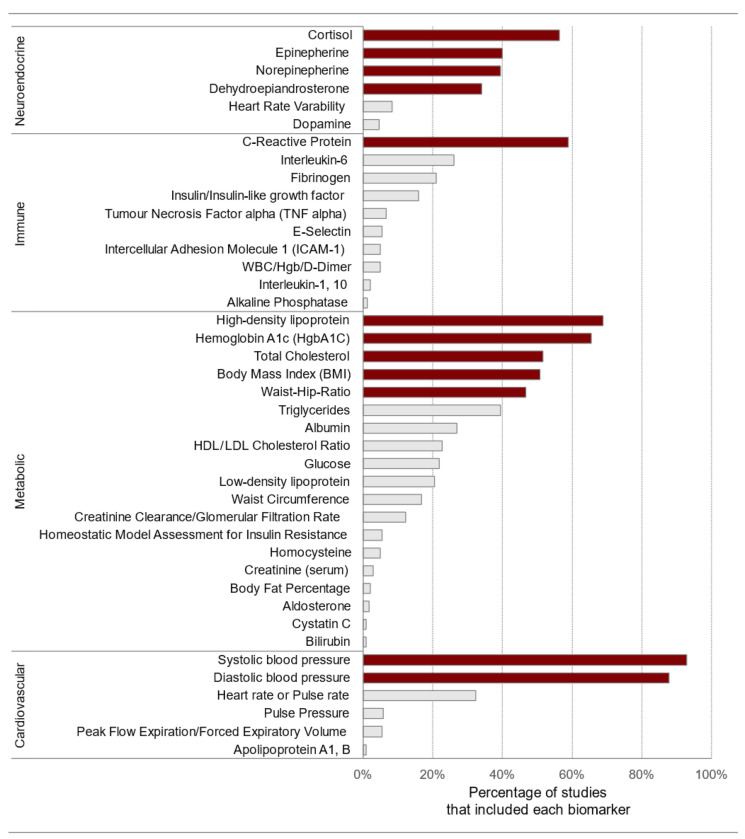
Frequency of biomarker use in the allostatic load index. The biomarkers that are shaded in a darker color include the initial ten variables, body mass index, and C-reactive protein. HgbA1C: glycosylated hemoglobin; WBC: white blood cell count; Hgb: hemoglobin. Source: reproduced with permission from Beese et al. (2022) [106].

**Figure 9 life-15-00867-f009:**
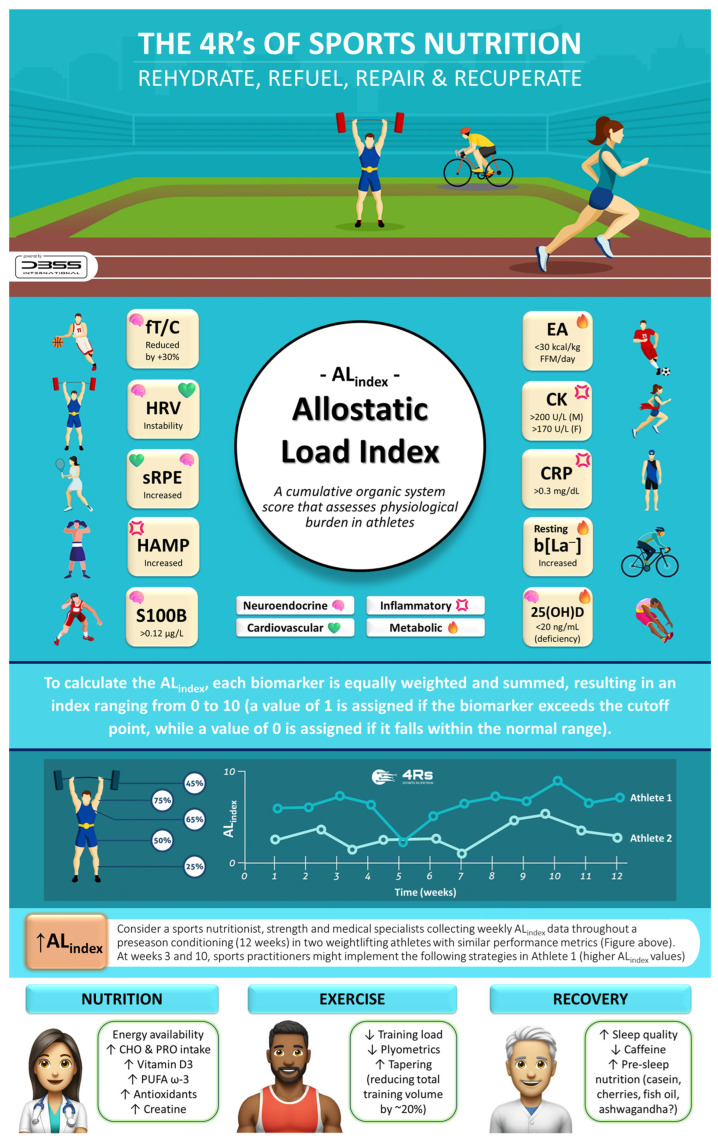
The 4Rs of sports nutrition and the allostatic load index in athletes. 25(OH)D: total serum 25-hydroxy vitamin D (i.e., the sum of D_3_ and D_2_); CHO: carbohydrates; CK: creatine kinase; CRP: C-reactive protein; EA: energy availability; F: female; FFM: fat-free mass; fT/C: free testosterone to cortisol ratio; HRV: heart rate variability; HAMP: hepcidin; b[La^−^]: blood lactate concentration; M: male; PRO: protein; PUFA ω-3: polyunsaturated fatty acids omega-3; S100B: S100 calcium-binding protein B; sRPE: session rating of perceived exertion. Source: designed by the authors (D.A.B.).

**Figure 10 life-15-00867-f010:**
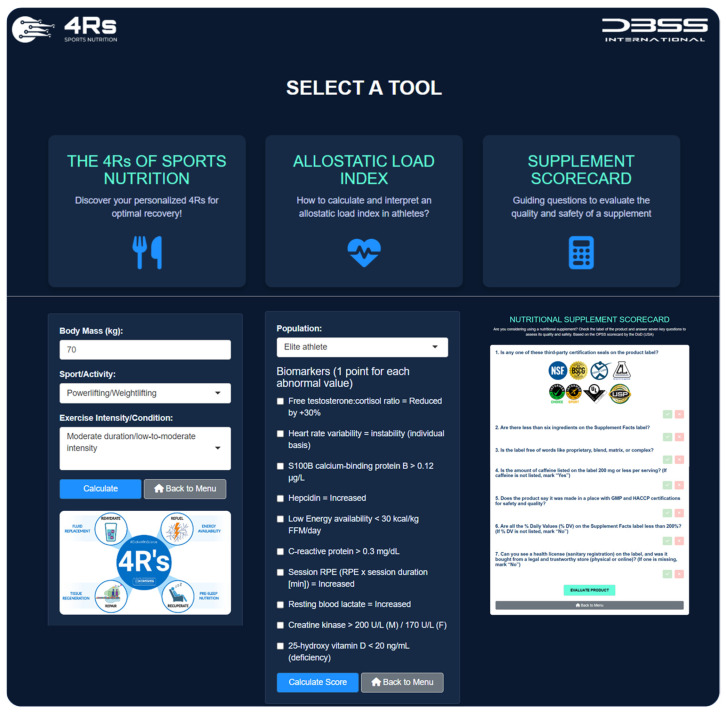
The 4Rs app. Available at https://dbss.shinyapps.io/4RsApp/ (accessed on 22 April 2025). Source: developed by D.A.B.

**Table 1 life-15-00867-t001:** Clinical evidence and recommendations by professional organizations on the 4Rs.

4Rs	Practical Recommendation	Clinical Evidence *	Professional Organization
Rehydration	Fluid replacement is a fundamental nutritional strategy that depends on the athlete’s needs, the environment, and the specific sports event. Adequate rehydration involves ~1.5 L per kg of body mass lost post-exercise, with electrolytes (mainly sodium) and carbohydrates (<6% *w*/*v*) to promote faster recovery.	López-Torres et al. (2023) [3] Pérez-Castillo et al. (2023) [4] Rowlands et al. (2022) [5] Zubac et al. (2019) [6] Holland et al. (2017) [7]	IOC (2023) [8] GSSI (2023) [9] UEFA (2021) [10] DGE (2020) [11] SDA (2020) [12] ISSN (2018) [13] NATA (2017) [14] ACSM (2007) [15]
Refuel	Carbohydrate intake (~1.2 g/kg body mass per hour for up to 4 h post-exercise) is essential not only in restoring glycogen reserves but also in supporting the energy needs of the immune system and facilitating tissue repair. Specifically, 20 g of creatine (5 g dose on four occasions beginning on the same day of fatiguing exercise) may promote muscle glycogen resynthesis in the first 24 h post-exercise. Despite changes in substrate utilization, a ketogenic diet generally has neutral or negative effects on athletic performance compared to carbohydrate-rich diets.	Cheng et al. (2025) [16] Lehman et al. (2024) [17] Ramos-Campo et al. (2024) [18] Díaz-Lara et al. (2024) [19] Koerich et al. (2023) [20] Craven et al. (2021) [21] Margolis et al. (2021) [22] Nielsen et al. (2020) [23] McCartney et al. (2018) [24] Roberts et al. (2016) [25]	ISSN (2024) [26] UEFA (2021) [10] DGE (2020) [27] DGE (2020) [28] ISSN (2018) [13] AND, DC, and ACSM (2016) [29]
Repair	Post-exercise ingestion of high-quality protein (0.3—0.5 g/kg body mass) and creatine monohydrate (0.1 g/kg body mass) supports tissue growth and repair. The potential of tart cherry, omega-3 fatty acids, dietary nitrate (e.g., *Beta vulgaris*, *Amaranthus* L.), and other herbal extracts containing flavonoid-rich polyphenols deserves further clinical research.	Pearson et al. (2023) [30] Doma et al. (2022) [31] Jones et al. (2022) [32] Hill et al. (2021) [33] Jiaming and Hossein (2021) [34] Carey et al. (2021) [35] Gao and Chilibeck (2020) [36] Morton et al. (2018) [37]	ISSN (2025) [38] DGE (2020) [39] ISSN (2017) [40] ISSN (2017) [41] AND, DC, and ACSM (2016) [29]
Rest/ Recuperate	Optimal sleeping time and quality are necessary to benefit the allostatic response after exercise. Alcohol should be avoided due to its inhibitory influence on several aspects of recovery. Ideally, caffeine should not be consumed for up to 4 h before bed. Pre-sleep nutrition has a restorative effect, facilitating the recovery of the musculoskeletal, endocrine, immune, and nervous systems. Pre-sleep nutritional strategies include whey, casein, or protein-rich meals; >150 mg of aqueous ashwagandha root extract; cherries; kiwi fruit; fish oils (omega-3 PUFAs); and valerian.	Gardiner et al. (2024a) [42] Gardiner et al. (2024b) [43] Gong et al. (2024) [44] Fatima et al. (2024) [45] Trommelen et al. (2023) [46] Reis et al. (2021) [47] Dela Cruz and Kahan (2021) [48] Cheah et al. (2021) [49] Gratwicke et al. (2021) [50] Walsh et al. (2021) [51] Snijders et al. (2019) [52] Barnes (2014) [53]	AASM (2021) [54] ISSN (2017) [55]

* Preference was given to meta-analytic evidence when available. ACSM: American College of Sports Medicine; AND: Academy of Nutrition and Dietetics; DC: Dietitians of Canada; DGE: Deutschen Gesellschaft für Ernährung—German Nutrition Society; GSSI: Gatorade Sports Science Institute; IOC: International Olympic Committee; NATA: National Athletic Trainers’ Association; PUFAs: polyunsaturated fatty acids; SDA: Sports Dietitians Australia; UEFA: Union of European Football Associations.

**Table 2 life-15-00867-t002:** Recommendations to evaluate body composition.

Method	Recommendations
Dual-Energy X-Ray Absorptiometry (DXA)	It is the gold-standard technique to measure bone content. Provides an estimation of fat mass (FM) and fat-free mass (FFM). Lean soft mass is estimated by subtracting bone minerals from FFM. Overall high reliability, although affected by nutrition/hydration status. FFM should be corrected for fat-free adipose tissue (FFAT). Certifications by the International Society for Clinical Densitometry.
Kinanthropometry	It measures skinfolds, girths, breadths, and lengths in trunk and upper and lower limbs. It represents a portable, fast, and more affordable method to track changes in body composition. The use of non-specific estimation equations for FM or FFM should be avoided. Absolute data analysis is preferred: sum of skinfolds and skinfold-corrected girths. Reliability depends on the technician’s experience and adherence to the ISAK standards. Certifications by the International Society for the Advancement of Kinanthropometry.
Bioelectrical Impedance Analysis (BIA)	It measures resistance (R) and reactance (Xc) at different frequencies (generally 50 kHz) of alternating current. Impedance (Z) is a vector sum of R and Xc. Estimates body composition components (FM, FFM, total body water, etc.) by integrating equations that use R, Xc, or Z as regressors. Bioelectrical impedance vectorial analysis is preferred for data analysis (phase angle and the Xc/R graph). Certifications by the International Society for Electrical Bioimpedance.
Air Displacement Plethysmography	It measures volume and mass to estimate body density based on air displacement. Provides estimations of FM, body fat percentage, and FFM by integrating equations that use body density as a regressor (e.g., Siri, Brozek). It does not provide regional data, and it is based on a two-compartment model that assumes fixed densities for adipose and musculoskeletal tissue. Relatively rapid technique with high test–retest reliability and useful for several populations, including special groups.

**Table 3 life-15-00867-t003:** Original and revised allostatic load indices for the general population.

The “Original 10” of AL_index_	Revised Version of AL_index_	Cut-Off Point	System
DHEA-S	Hair cortisol	>150 pg/mg	Neuroendocrine
Cortisol	C-reactive protein *	≥0.3 mg/dL	Inflammatory
Epinephrine	Resting heart rate *	≥90 beats/min	Cardiovascular
Norepinephrine	Systolic blood pressure	≥140 mmHg	
Systolic blood pressure	Diastolic blood pressure	≥90 mmHg	
Diastolic blood pressure	HDL cholesterol *	<40 mg/dL (men); <50 mg/dL (women)	
HDL cholesterol	Total cholesterol	≥240 mg/dL	
Total cholesterol	Serum albumin	<3.8 g/dL	Metabolic
HbA1c	HbA1c *	≥6.4%	
Waist-to-hip ratio	Waist-to-height ratio *	>0.5	

A new proposal to evaluate allostatic load in populations with chronic diseases. All blood markers should be assessed in a rested and fasted condition. * Recent meta-analytic evidence suggests that a five-item AL_index_, based on C-reactive protein, resting heart rate, HDL cholesterol, waist-to-height ratio, and HbA1c, can predict mortality as effectively as, or better than, more complex biomarker sets [107]. DHEA-S: dehydroepiandrosterone sulfate; HbA1c: glycosylated hemoglobin; HDL: high-density lipoprotein.

## Data Availability

The data supporting this review are from previously reported studies and datasets, which have been cited. The CAT2 tool is available at https://stillmed.olympics.com/media/Documents/Athletes/Medical-Scientific/Consensus-Statements/REDs/IOC-REDs-CAT-V2.pdf (accessed on 20 August 2024). We encourage readers to explore the blogs and whitepapers of the ALLOSTASIS—Decentralized Autonomous Organization (https://www.allostasis.io/, accessed on 28 November 2024) and VERSES, a cognitive computing company building next-generation intelligent software systems (https://www.verses.ai/, accessed on 28 November 2024), as these initiatives offer insights and scientific literacy regarding allostasis and the free energy principle.

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
