# Peer review of "The 4Rs Framework of Sports Nutrition: An Update with Recommendations to Evaluate Allostatic Load in Athletes"

_life, 2025, doi:10.3390/life15060867_

Round 1

Reviewer 1 Report

Comments and Suggestions for Authors

Overall

The manuscript presents an updated perspective on the 4R’s framework in sports nutrition, emphasizing the role of allostasis in athlete recovery and adaptation. The topic is timely and relevant, given the increasing integration of sports science and data analytics in optimizing athletic performance.

The manuscript is well-researched, thoroughly referenced, and structured logically.

However, some areas require further clarification, refinement, and critical analysis.

   - The paper is well-grounded in existing literature, with an extensive reference list supporting the claims made. The authors should be applauded for these efforts.

   - The application of allostatic load assessment to sports nutrition is innovative and provides a novel perspective that could have practical implications for performance optimization. As the field of athlete preparedness is becoming more normative in practice, this research may bridge the gap between the worlds of academia and sport performance/athlete health.

   - The structured breakdown of the 4R’s (Rehydrate, Refuel, Repair, and Recuperate) makes the paper accessible to both researchers and practitioners in strength and conditioning. This formatting does allow for quick digestibility of the information upon initial exposure to the manuscript, which is backed by the well research written information.

   - The inclusion of practical recommendations, clinical applications, and biomarker assessments makes this work applicable to real-world settings. As the work is centered on practical application, this aspect is paramount.

 Areas for Consideration

   - The ALindex is a central concept in this manuscript, but its explanation could be made clearer, particularly in terms of how it is calculated, validated, and interpreted in an applied sports setting. Please check this aspect of the paper to insure there are no gaps in explanation. As one becomes more familiar with the content of an article, we sometimes lose the perspective of the reader.

   - Providing a worked example or case study illustrating how an athlete's allostatic load is assessed and managed over time would enhance comprehension. This may be an expansion of what you have already provided. However, if you believe it warranted a practical example could enhance the manuscript. (merely a point of consideration)

   - While the paper effectively explains the physiological rationale behind the 4R’s, practical implementation strategies (e.g., real-world training plans, sport-specific considerations) could be further elaborated. Realizing this may be a difficult ask as 4R’s is ultimately individualized and this comment echoing the suggestion made above.

   - Providing concrete guidelines on how coaches and sports scientists can integrate ALindex into training and nutrition programming would make the recommendations more actionable. You do address this a bit in the Future Directions section. Perchance this would an article moving forward.

   - The manuscript does not adequately discuss the potential limitations of the 4R’s framework and ALindex, such as inter-individual variability, reliability of biomarkers, and practical constraints in real-world settings. Some of these are mentioned in the Future Directions section but are not listed as potential limitations.

   - Expanding on gaps in research and potential next steps, including how ALindex might be standardized across different populations, would enhance the manuscript’s contribution.

  - Ensure consistent citation formatting (e.g., some references lack uniformity in style). This is in reference to the inclusion of DOI and URL in some of the citations and not all. Please check this to reach a greater level of consistency.

The manuscript is well-researched and presents an important contribution to the field of sports nutrition. However, it requires refinements.

Expanding the discussion on limitations and implementation strategies will further enhance its impact. 

Author Response

Reviewer #1

The manuscript presents an updated perspective on the 4R’s framework in sports nutrition, emphasizing the role of allostasis in athlete recovery and adaptation. The topic is timely and relevant, given the increasing integration of sports science and data analytics in optimizing athletic performance.

The manuscript is well-researched, thoroughly referenced, and structured logically.

However, some areas require further clarification, refinement, and critical analysis.

   - The paper is well-grounded in existing literature, with an extensive reference list supporting the claims made. The authors should be applauded for these efforts.

   - The application of allostatic load assessment to sports nutrition is innovative and provides a novel perspective that could have practical implications for performance optimization. As the field of athlete preparedness is becoming more normative in practice, this research may bridge the gap between the worlds of academia and sport performance/athlete health.

   - The structured breakdown of the 4R’s (Rehydrate, Refuel, Repair, and Recuperate) makes the paper accessible to both researchers and practitioners in strength and conditioning. This formatting does allow for quick digestibility of the information upon initial exposure to the manuscript, which is backed by the well research written information.

   - The inclusion of practical recommendations, clinical applications, and biomarker assessments makes this work applicable to real-world settings. As the work is centered on practical application, this aspect is paramount.

Response: We appreciate the reviewer’s comments. Several amendments have been performed to improve the manuscript. Point-by-point answers for each reviewer’s comment have been created. We have also included the requested modifications in the revised manuscript. All changes have been clearly highlighted in red (tracking changes) so that they can be easily visible to the editor and reviewers.

 Areas for Consideration

   - The ALindex is a central concept in this manuscript, but its explanation could be made clearer, particularly in terms of how it is calculated, validated, and interpreted in an applied sports setting. Please check this aspect of the paper to insure there are no gaps in explanation. As one becomes more familiar with the content of an article, we sometimes lose the perspective of the reader.

Response: Thanks for your comment. We have revised the manuscript accordingly. See changes in section 5. How to measure the Allostatic Load in athletes? (Page 15), Table 4 reformatting, and Section 6. Applied practice. 

   - Providing a worked example or case study illustrating how an athlete's allostatic load is assessed and managed over time would enhance comprehension. This may be an expansion of what you have already provided. However, if you believe it warranted a practical example could enhance the manuscript. (merely a point of consideration)

   - While the paper effectively explains the physiological rationale behind the 4R’s, practical implementation strategies (e.g., real-world training plans, sport-specific considerations) could be further elaborated. Realizing this may be a difficult ask as 4R’s is ultimately individualized and this comment echoing the suggestion made above.

Response: Thank you for your suggestion. We have revised the manuscript to enhance the details on biomarker selection, scoring algorithm, and the relevant discussions on the topic.

   - Providing concrete guidelines on how coaches and sports scientists can integrate ALindex into training and nutrition programming would make the recommendations more actionable. You do address this a bit in the Future Directions section. Perchance this would an article moving forward.

   - Expanding on gaps in research and potential next steps, including how ALindex might be standardized across different populations, would enhance the manuscript’s contribution.

Response: Thank you for your suggestion. Further clinical evidence is required before establishing guidelines on this topic (e.g., GRADE systematic reviews with meta-analysis or Delphi-based consensus). We have revised the text to include information about the ALindex scoring algorithm, its foundations, and gaps that will guide future research.

   - The manuscript does not adequately discuss the potential limitations of the 4R’s framework and ALindex, such as inter-individual variability, reliability of biomarkers, and practical constraints in real-world settings. Some of these are mentioned in the Future Directions section but are not listed as potential limitations.

Response: Thanks for your suggestion. We have added limitations where suitable (lines 541-575).

  - Ensure consistent citation formatting (e.g., some references lack uniformity in style). This is in reference to the inclusion of DOI and URL in some of the citations and not all. Please check this to reach a greater level of consistency.

Response: Thank you for your comment. We have reviewed and revised the references to ensure consistency. The inclusion of URLs in some citations is intentional, as they correspond to databases, technical reports, or other non-article documents, which, according to MDPI reference style, require this format.

The manuscript is well-researched and presents an important contribution to the field of sports nutrition. However, it requires refinements. Expanding the discussion on limitations and implementation strategies will further enhance its impact.

Response: We appreciate all the comments and suggestions from the reviewer. We have revised the manuscript based on the feedback from all reviewers and are pleased with the improved version. We hope the reviewers feel the same.

Reviewer 2 Report

Comments and Suggestions for Authors

-Conceptual and Theoretical Weaknesses

-The paper strongly advocates allostasis as a superior paradigm over homeostasis but does not critically evaluate its limitations.

-Many aspects of exercise recovery (e.g., muscle protein synthesis, glycogen restoration) are well explained within homeostatic models, making the exclusive use of allostasis questionable.

-A balanced discussion contrasting homeostasis vs. allostasis in sports science would strengthen the theoretical foundation.

-The ALindex is presented as a novel tool for monitoring athletes and validation studies are missing: There is no empirical evidence (longitudinal or interventional) showing that the proposed biomarkers accurately predict performance or recovery. The weighting of each biomarker is arbitrary, assuming equal physiological impact, which is unlikely. Cut-off values for biomarkers (e.g., creatine kinase, heart rate variability) are not well justified and may lack specificity for elite athletes.

-Comparing ALindex to existing validated recovery scores (e.g., HRV, cortisol variability, lactate thresholds) would improve credibility.

-Lack of Experimental or Longitudinal Data

-The article summarizes existing research but does not provide new experimental data.

-The ALindex proposal lacks validation in real-world settings: There is no proof that tracking these biomarkers improves training outcomes.

-Search Strategy in PRISMA-ScR Is Superficial

-Selection criteria for included studies.

-A table summarizing study quality, level of evidence, and risk of bias would improve

-The ALindex Is Impractical for Many Athletes: Frequent biomarker testing is costly and inaccessible. Interpretation of biomarkers requires expertise, making it impractical for self-guided athletes.

Therefore, simplified version with non-invasive markers (e.g., HRV, resting heart rate, perceived fatigue) would be more feasible.

-Overemphasis on Biomarkers Without Contextualizing Individual Variation

- Biomarkers should not be interpreted in isolation, yet the paper does not propose a decision-making algorithm.

-Therefore,  personalized ranges based on athlete history and integration with subjective fatigue questionnaires would improve applicability.

-Ethical and Conflicts of Interest Concerns

-Several authors have industry affiliations with sports supplement companies.

-The funding section mentions sponsorship from Texas A&M's Exercise & Sport Nutrition Laboratory (ESNL), which has ties to the dietary supplement industry.

-The article recommends multiple supplements (e.g., creatine, omega-3, tart cherry) without sufficient critical discussion on their necessity. A clearer conflict-of-interest statement and a more critical evaluation of supplementation research would improve credibility. Address conflicts of interest: Clearly distinguish evidence-based recommendations from industry influence.

Author Response

Dear Reviewer,

Thank you for your comments and suggestions.
Please find the point-by-point response in the attached file.

We look forward to your feedback.

Sincerely,
The Authors

Reviewer 3 Report

Comments and Suggestions for Authors

This study examines a new paradigm for nutritional management of the post-exercise recovery process, including chronic stress management, and is expected to be of great help in conditioning management for athletes.

The research topic in the introduction and research methods are also well described. In addition, the pictures are organized so that the overall content can be easily understood.

One thing that is disappointing is that I wish the nutritional program related to allostatic load could be presented in more detail.

Author Response

Reviewer #3

This study examines a new paradigm for nutritional management of the post-exercise recovery process, including chronic stress management, and is expected to be of great help in conditioning management for athletes.

The research topic in the introduction and research methods are also well described. In addition, the pictures are organized so that the overall content can be easily understood.

One thing that is disappointing is that I wish the nutritional program related to allostatic load could be presented in more detail.

Response: We appreciate the reviewer’s comments. Several amendments have been made to improve the manuscript based on the feedback and suggestions provided by the evaluators. It is important to emphasize that the allostatic load index should be viewed as a multisystem tool for monitoring and adjusting the exercise, nutrition, and sleep program based on the athlete’s responses during preparation or competition. Individualization and adherence to the program are crucial, as highlighted in the manuscript.

Round 2

Reviewer 2 Report

Comments and Suggestions for Authors

Your work presents a timely and innovative synthesis of nutritional recovery strategies grounded in the concept of allostasis. The integration of the 4R framework with a proposed Allostatic Load Index (ALindex) offers a valuable theoretical model for understanding and managing recovery in athletic populations.

Nevertheless, it is recommended that the manuscript undergoes further refinement to ensure effective translation into practice, particularly with regard to accounting for inter-individual variability among athletes.

Key Points for Revision: Adaptation of the 4R Framework to Athlete Profiles

At present, the 4R recommendations (Rehydrate, Refuel, Repair, Recuperate) are presented in a non-specific format. However, effective application must consider the athlete's individual context, including:

- Sport type (e.g., aerobic vs. anaerobic demand)

- Training phase or load

- Sex, paying particular attention to menstrual cycle phase in female athletes

- Age, which influences anabolic resistance, inflammatory responses, and recovery kinetics

- Energy availability, especially in relation to RED-S

For example, carbohydrate needs, protein timing, or pre-sleep recovery strategies will not be the same for a female endurance runner in the luteal phase as for a male strength athlete in taper week. In order to enhance the practical relevance and ethical scope of the recommendations under discussion, it is recommended that they include sex-specific and hormonal-cycle-specific adjustments.

The refinement of the Allostatic Load Index with greater individualisation is a commendable endeavour. However, the proposed index is compelling in its structure but does not yet account for individualised cut-offs or athlete profiling by sport, age, or sex. While the article acknowledges the limitations of current biomarker strategies, it does not offer practical solutions for tailoring biomarker thresholds or recovery strategies.

A notable omission is the absence of menstrual-related markers or energy deficiency-related hormonal changes (e.g., progesterone, estradiol, leptin) in female athletes, which is a critical gap in a tool aiming to support physiological adaptation and load management.

The article would be strengthened by the provision of decision trees, case examples, or conditional logic (e.g., if/then scenarios) to assist practitioners in adjusting each R. Suggestions might include:

- Methodologies for periodising carbohydrate or protein intake in accordance with fatigue markers or biomarker feedback.- Interpretive approaches to fluctuations in the ALindex in relation to performance trends or injury risk.- Adaptation of recovery strategies in consideration of hormonal contraceptive use or menstrual phase.

Strengths Acknowledged

The manuscript is characterised by a robust structure and extensive referencing. The integration of concepts such as allostasis, interoception, and RED-S, as well as the call for multidimensional monitoring, reflects a high level of theoretical sophistication and scientific maturity.

In summary, it is strongly recommended that the manuscript be revised to incorporate athlete-specific adaptations and sex-based physiological differences, particularly in the case of female athletes. Incorporating cycle-phase-sensitive nutritional strategies, sport-specific adaptations, and dynamic use of the ALindex will enhance the framework's impact and applicability across diverse populations.

It is believed that the work has the potential to become a reference in the field, particularly if these refinements are made.

Author Response

Dear Reviewer,

Thank you for your comments and suggestions.
Please find the point-by-point response in the attached file.

Sincerely,
The Authors